# Influence of Sulphur Content on Structuring Dynamics during Nanosecond Pulsed Direct Laser Interference Patterning

**DOI:** 10.3390/nano11040855

**Published:** 2021-03-27

**Authors:** Theresa Jähnig, Cornelius Demuth, Andrés Fabián Lasagni

**Affiliations:** 1Institute of Manufacturing Technology, Technische Universität Dresden, P.O. Box, 01062 Dresden, Germany; cornelius.demuth1@mailbox.tu-dresden.de; 2Fraunhofer Institute for Material and Beam Technology IWS, Winterbergstr. 28, 01277 Dresden, Germany

**Keywords:** direct laser interference patterning, periodic microstructure, sulphur content, nanosecond pulse, surface tension gradient, Marangoni convection, smoothed particle hydrodynamics

## Abstract

The formation of melt and its spread in materials is the focus of many high temperature processes, for example, in laser welding and cutting. Surface active elements alter the surface tension gradient and therefore influence melt penetration depth and pool width. This study describes the application of direct laser interference patterning (DLIP) for structuring steel surfaces with diverse contents of the surface active element sulphur, which affects the melt convection pattern and the pool shape during the process. The laser fluence used is varied to analyse the different topographic features that can be produced depending on the absorbed laser intensity and the sulphur concentration. The results show that single peak geometries can be produced on substrates with sulphur contents lower than 300 ppm, while structures with split peaks form on higher sulphur content steels. The peak formation is explained using related conceptions of thermocapillary convection in weld pools. Numerical simulations based on a smoothed particle hydrodynamics (SPH) model are employed to further investigate the influence of the sulphur content in steel on the melt pool convection during nanosecond single-pulsed DLIP.

## 1. Introduction

During the process of evolution, numerous animals and plants have perfectly adapted to environmental conditions by developing repetitive surface patterns. For instance, fine riblets on their skin scales, which reduce water drag, enable sharks to swim faster and more effortlessly than other fishes [1]. In addition, hierarchical structures give rise to the structural colour of *Morpho* butterfly wings and to the hydrophobic and self-cleaning properties of rice leaves [2,3]. Learning lessons from nature and imitating these examples, surface microstructuring can be applied to improve the efficiency of ship propulsion and electricity generation in wind turbines by reducing friction, and to generate colours for decorative motifs or security marks on products as well as self-cleaning surfaces on solar panels or worktops in professional kitchens [4,5]. The success of manufacturers in a global competition relies on additional benefits to differentiate their products. Consequently, techniques allowing for the fabrication of complex surface patterns are sought to add novel functionalities to products.

Surface topographies with feature sizes in a wide range can be generated, from submicron structures made by lithography and subsequent wet chemical etching [6] or focussed beam structuring [7] to submillimetre surface textures produced by brushing, grinding, or milling. After structuring, several cleaning steps are, in general, required before the surface can be used further, with the extended production time impairing the profitability of the overall process.

In contrast, laser texturing is an effective single-step method for fabricating surface structures that dispenses with the need for material removal or cleaning processes [8,9]. More specifically, direct laser interference patterning (DLIP) provides the advantage of producing repetitive surface structures with periodicities in the submicron to micron range. This surface functionalisation technique has already been employed for diverse applications, such as antibacterial and antifouling surfaces [10], control of wettability [11], tribology [12,13], cell adhesion [14,15], and decoration and anti-counterfeit protection [16] on various materials. The DLIP process generates periodic microstructures on substrates exposed to an interference pattern, which results when at least two coherent laser beams are combined, by means of photothermal or photochemical interactions [17,18,19].

The intensity distribution of the interference pattern is determined by the laser wavelength λ, the number and the polar angle θ/2 of the partial beams [20,21]. A two-beam configuration considered here generates line-like surface textures, where the spatial period Λ, defined as the distance between adjacent intensity maxima or minima, can be adjusted by the intersecting angle θ according to Equation (Equation 1): (1)$$\Lambda =\frac{\lambda }{2\sin\left(\theta /2\right)}.$$
It is evident from Equation (Equation 1) that the minimal theoretically achievable spatial period is half of the used laser wavelength, i.e., Λ=λ/2 for an interception angle of θ=π. The sinusoidal laser fluence distribution of a two-beam interference pattern is given by Equation (Equation 2): (2)$$\Phi \left(x,y\right)=4\Phi_{0}\cos^{2}\left[kx\sin\left(\frac{\theta }{2}\right)\right],$$
where $\Phi_{0}$ is the fluence of each partial beam and the wave number is defined as k=2π/λ.

The application of short pulsed laser radiation with nanosecond pulse duration and infrared wavelengths involves photothermal interactions, in particular the formation of very localised molten zones [22,23]. In the case of metallic substrates, the local insertion of energy in a thin surface layer enables the specific shaping of well-defined melt pools associated with an alteration of the surface topography, which is influenced by the laser fluence and the material properties as well. In nanosecond pulsed DLIP of metals at moderate laser fluence, thermocapillary driven melt pool flow is interpreted as the mechanism responsible for the surface microstructure formation [24]. The in-process observation of DLIP is not possible though, owing to the short duration and the microscopic scale of the surface modification. However, the use of numerical simulation allows for an investigation and insights into the phenomena contributing to the process.

The development of the melt pool convection was originally modelled for welding processes with continuous energy input into the metallic workpiece. In detail, arc welding was considered at first, and the research was subsequently extended to laser beam welding. Accordingly, the melt pool characteristics are determined by the interaction between arc and metal, the heat transfer and fluid flow along with the thermophysical material properties, and the applied boundary conditions [25,26,27,28]. Numerical investigations on thermal fluid flow revealed that thermocapillary stresses constitute the main force driving the circulation in the weld pool [27,28]. This thermocapillary flow, or Marangoni convection, arises due to surface tension gradients along the melt pool surface [29]. Even minor additions of a surface active element, e.g., oxygen, aluminium, phosphorus, or sulphur, may significantly affect the surface tension of molten metal and, as a result, the weld penetration depth and pool width [30,31,32].

The effect of surface active elements on the melt convection patterns and pool shape was first investigated experimentally by Heiple and Roper [29]. Beginning at that time, mathematical models for the theoretical study of welding processes, i.e., the fluid flow [30] and heat transfer in the melt pool [33,34,35,36,37] were developed. While the effect of the temperature-dependent surface tension was incorporated into the models [33,34,35,36,37], the weld pool simulations mostly considered a constant temperature coefficient of surface tension. On the other hand, a variable, i.e., a temperature-dependent, temperature derivative of surface tension, as required in the presence of a surfactant, was rarely included in the numerical investigations [38,39,40,41,42].

A very low sulphur content in steel (<30 ppm or at most 40 ppm S) involves a negative temperature coefficient of surface tension [29,32], giving rise to an outward melt flow at the surface and a wide but shallow weld pool. For a higher sulphur content in steel, the temperature derivative of surface tension is positive over a temperature interval of at least 200 K from the melting point [29]. A completely inward convection at the surface and a narrow but deep weld pool result [42], provided that the surface temperature does not exceed the aforementioned range. Furthermore, thermocapillary convection is assumed to influence the surface shape of the fusion zone [32,43]. Outward thermocapillary flow leads to a recess at the centre and bulges near the edges of the melt pool surface, as outlined in Figure 1a. On the contrary, inward Marangoni convection entails a ridge at the centre and surface depressions adjacent to the edges of the molten zone; see Figure 1b.

Notwithstanding the extensive research on thermocapillary flow in the context of welding, the effects of surface active elements in steel on the mechanisms of laser-based microprocessing are still widely unexplored. With regard to laser texturing, the melt pool flow may be employed to generate novel microstructures that provide surfaces with particular functionalities.

In this work, steel samples with different sulphur content are treated by two-beam direct laser interference patterning using a nanosecond pulsed laser to explore the possibilities of fabricating repetitive surface patterns based on thermocapillary convection due to the negative or positive temperature coefficient of surface tension. The topographies resulting from the two-beam DLIP process are analysed by confocal and scanning electron microscopy. Depending on the employed laser processing parameters and the sulphur content, individual structuring regimes are identified. In addition, numerical process simulations are performed using a smoothed particle hydrodynamics (SPH) model to understand the thermocapillary flow patterns in the melt pool and hence the microstructure formation.

## 2. Materials and Methods

This section presents the techniques and models employed in the combined experimental and numerical investigation. At first, the steel substrates used, the experimental setup, and the sample characterisation methods are detailed in Section 2.1–Section 2.3. Following this, the theoretical model and the specific parameters considered for the numerical simulations are elucidated in Section 2.4 and Section 2.5.

### 2.1. Materials

In order to evaluate the influence of the sulphur content (S) on the structure formation with DLIP, steels with four different sulphur contents were chosen: 30 ppm, 100 ppm, <300 ppm, and 1500–3000 ppm S. The flat steel plates with a thickness of 5 mm were cut into 50 × 50 mm^2^ pieces and polished with diamond suspension (Masterprep, Buehler, Lake Bluff, IL, USA) of 0.05 µm size to an average surface roughness (*Sa*) of 0.008 µm. The root mean square roughness (*Sq*) of the same samples was 0.020 µm, with a maximum height of the assessed profile (*Sz*) at 0.450 µm. Before structuring, the surfaces were cleaned in an ultrasonic bath with ethanol (99.99% purity) for 25 min at room temperature to remove all preprocess contamination and were dried with compressed air.

### 2.2. Nanosecond Direct Laser Interference Patterning

The laser structuring process was accomplished on a compact DLIP system (DLIP-µFab, Fraunhofer IWS, Dresden, Germany) equipped with a Q-switched Nd:YLF laser (Tech-1053 Basic, Laser-export, Moscow, Russia) operating at 1053 nm and providing 12 ns pulses (full width at half maximum, FWHM) at 1 kHz with pulse energies up to 290 µJ. The laser emits the fundamental transverse mode (TEM${}_{00}$) with a laser beam quality factor of M^2^ < 1.2. This main beam is then split into two sub-beams by passing through a diffractive optical element (DOE) and the beams are subsequently parallelised by a prism. A schematic representation of the main DLIP components is shown in Figure 2a. A lens is overlapping the beams on the surface with a focal distance of 40 mm and produces an interference spot of 160 µm in diameter. Varying the angle of incidence of the partial beams by changing the distance between DOE and prism, the spatial period determined by Equation (Equation 1) is specified in the range between 1.29 µm and 7.20 µm. The latter period was chosen for the structuring on the steel samples to keep the single line-like melt pools separated from each other and to prevent thermal influence among themselves.

The Gaussian shape of the intensity profile of the overlapped beams is indicated in Figure 2b. The highest laser intensity is reached in the spot centre, whereas the intensity decreases towards the spot edges. The DLIP process was performed using laser fluences between 0.2 J/cm^2^ and 1.1 J/cm^2^, which correspond to 58% to 91% of laser power, respectively. The samples were positioned under the laser spot with an *x*-*y*-axis system (Pro115 linear stages, Aerotech Inc., Pittsburgh, PA, USA), and the focus position was controlled by vertically moving the DLIP optical head on a *z*-axis. Laser microprocessing experiments were carried out in ambient environment without posttreatments. In all cases, the structuring process was made without overlapping pulse.

Here, the angle of incidence θ/2 of the partial beams is considered small and, as a consequence, the interference spot is assumed to be circular. Therefore, the laser fluence distribution of the interference pattern exemplified in Figure 2b is given by
(3)$$\Phi \left(x,y\right)=2\Phi_{0}\left\{\cos\left[2kx\sin\left(\frac{\theta }{2}\right)\right]+1\right\}\exp\left(-\frac{2r^{2}}{r_{0}^{2}}\right),$$
where $r^{2}=x^{2}+y^{2}$ and $r_{0}$ is the Gaussian beam radius of the partial beams. Assuming that the periodicity Λ (see Equation (Equation 1)) of the interference pattern is considerably smaller than the Gaussian beam diameter, the average fluence is given by
(4)$$\Phi_{\mathrm{av}}\left(x,y\right)=2\Phi_{0}\exp\left(-\frac{2r^{2}}{r_{0}^{2}}\right)$$
with the central value $\Phi_{\mathrm{av}}\left(r=0\right)=2\Phi_{0}$. Furthermore, the interference spot radius $r_{\mathrm{spot}}$ is supposed to be smaller than the Gaussian beam radius $r_{0}$.

Consequently, the energy provided by the laser pulse is calculated by integrating the local averaged fluence in Equation (Equation 4) over the interference spot
(5)$$E_{\mathrm{spot}}=\int_{\phi =0}^{2\pi }\int_{r=0}^{r_{\mathrm{spot}}}\Phi_{\mathrm{av}}\left(r\right)r\;dr\;d\phi =\Phi_{0}\pi r_{0}^{2}\left[1-\exp\left(-\frac{2r_{\mathrm{spot}}^{2}}{r_{0}^{2}}\right)\right]$$
and divided by the spot area to obtain the fluence averaged over the interference spot
(6)$$\Phi_{\mathrm{av},\mathrm{spot}}=\frac{E_{\mathrm{spot}}}{\pi r_{\mathrm{spot}}^{2}}=\Phi_{0}\frac{r_{0}^{2}}{r_{\mathrm{spot}}^{2}}\left[1-\exp\left(-\frac{2r_{\mathrm{spot}}^{2}}{r_{0}^{2}}\right)\right].$$
Given that the averaged fluence $\Phi_{\mathrm{av},\mathrm{spot}}$ in Equation (Equation 6) is known from measurement, the average fluence in the centre of the spot is determined by
(7)$$\Phi_{\mathrm{av}}\left(r=0\right)=2\Phi_{0}=2\Phi_{\mathrm{av},\mathrm{spot}}\frac{r_{\mathrm{spot}}^{2}}{r_{0}^{2}}/\left[1-\exp\left(-\frac{2r_{\mathrm{spot}}^{2}}{r_{0}^{2}}\right)\right].$$
As mentioned above, it is assumed that $\Lambda /2=\lambda /\left(4\sin\left(\theta /2\right)\right)\ll r_{0}$. Therefore, the laser fluence distribution of the period comprising the central maximum and in the central plane, i.e., for |x|≤Λ/2 and y=0, is approximately given by Equation (Equation 2), which corresponds to Equation (Equation 3) without the exponential factor.

### 2.3. Surface Characterisation

The polished and structured surfaces were characterised using confocal microscopy (S neox, Sensofar, Terrassa, Spain) at 150× magnification with an optical resolution of 0.14 µm and a vertical resolution of 1 nm. The field of view (FOV) is 117 × 88 µm^2^ in the *x*-*y*-plane. The recorded measurements were processed using the software MountainsMap 7.3 (Digital Surf, Besançon, France), utilising profile extraction including the step height measurement function for analysis of the peak formation and the image tools for three-dimensional images of the structured surfaces. In addition, the laser treated substrates were also evaluated using a scanning electron microscope (SEM) (Supra 40 VP, Zeiss, Jena, Germany) at an operating voltage of 5.0 kV.

### 2.4. Mathematical Model

The mathematical model of the material behaviour during single pulse DLIP was already described in detail by Demuth and Lasagni [44]. Here, the nonlinear temperature dependence of surface tension considered is presented along with the dimensionless form of the governing equations and the Marangoni boundary condition. The dimensional conservation equations and boundary conditions are provided in Appendix A for the sake of clarity.

In the present investigation, the surface tension of liquid steel is a nonlinear function of temperature in the presence of the surface active element sulphur. Therefore, the temperature coefficient of surface tension, i.e., dγ/dT, is a function of temperature as well, unlike the constant negative value found in the absence of a surfactant.

An expression for the temperature-dependent surface tension of liquid iron with the solute sulphur was derived by Sahoo et al. [45] in the following form: (8)$$\gamma =\gamma_{m}^{0}+{\left.\frac{\;d\gamma }{\;dT}\right|}_{\begin{array}{c} \mathrm{pure}\\ \mathrm{metal} \end{array}}\left(T-T_{m}\right)-RT\Gamma_{s}\ln\left[1+k_{1}a_{S}\exp\left(-\frac{\Delta H_{0}}{RT}\right)\right],$$
where $\gamma_{m}^{0}$ is the surface tension of the pure metal at the melting point $T_{m}$ and $a_{S}$ is the sulphur activity in wt%. In Equation (Equation 8), R=8.3143 J/(molK) is the gas constant, $\Gamma_{s}=1.3\times 10^{-5}$ mol/m^2^ is the surface excess at saturation, $k_{1}=3.18\times 10^{-3}$ is a constant related to the entropy of segregation, and $\Delta H_{0}=-1.88\times 10^{5}$ J/mol is the standard enthalpy of adsorption [38,45]. The values of the temperature coefficient of surface tension, which is required for the melt pool convection in the boundary condition (Equation 29) (see Appendix A) resulting for the binary system Fe–S can be used for the simulation of stainless steel melt pools [46]. Further considering the values $\gamma_{m}^{0}=1.943$ N/m, ${\left.\;d\gamma /\;dT\right|}_{\mathrm{pure}\mathrm{metal}}=-4.3\times 10^{-4}$ N/(mK) and $T_{m}=1723$ K [38,45,47] in Equation (Equation 8), the surface tension is evaluated and presented in Figure 3a for liquid steel with different concentrations of the surfactant sulphur. Differentiating Equation (Equation 8) with respect to *T*, the relation for the temperature coefficient of surface tension is obtained as follows [38,45]: (9)$$\frac{\;d\gamma }{\;dT}={\left.\frac{\;d\gamma }{\;dT}\right|}_{\begin{array}{c} \mathrm{pure}\\ \mathrm{metal} \end{array}}-R\Gamma_{s}\ln\left(1+Ka_{S}\right)-\frac{Ka_{S}}{1+Ka_{S}}\frac{\Gamma_{s}\Delta H_{0}}{T},$$
where the equilibrium constant $K=k_{1}\exp\left[-\Delta H_{0}/\left(RT\right)\right]$ is introduced. Consequently, the temperature derivative of surface tension in Equation (Equation 9) is evaluated as a function of temperature for molten stainless steel with different sulphur contents and presented in Figure 3b.

The governing equations stated in (Equation 23)–(Equation 26) are rewritten in dimensionless form to reveal the dimensionless numbers characterising the problem. For this purpose, the non-dimensionalisation of the variables position x, time *t*, velocity v, dynamic pressure $p_{\mathrm{dyn}}$, temperature *T*, and specific enthalpy *h* is made using the scales *L*, $L^{2}/a$, a/L, $\rho_{0}a^{2}/L^{2}$, $T_{v}-T_{0}$, and $c_{p}\left(T_{v}-T_{0}\right)$, respectively. In addition, the diffusion length $L=2\sqrt{a\tau_{p}}$ is specified as the characteristic length scale, where the pulse width $\tau_{p}$ (FWHM) is selected as the laser beam dwell time. The resulting dimensionless variables are given by
(10)$$x^{*}=\frac{x}{2\sqrt{a\tau_{p}}},\;t^{*}=\frac{t}{4\tau_{p}},\;v^{*}=2v\sqrt{\frac{\tau_{p}}{a}},\;p_{\mathrm{dyn}}^{*}=\frac{4\tau_{p}}{\rho_{0}a}p_{\mathrm{dyn}},\;T^{*}=\frac{T-T_{0}}{T_{v}-T_{0}}$$
and the dimensionless specific enthalpy assumes the shape
(11)$$h^{*}=\frac{h}{c_{p}\left(T_{v}-T_{0}\right)}=T^{*}+f_{m}\;{\mathrm{Ph}}_{s/l}+f_{v}\;{\mathrm{Ph}}_{l/v},$$
where the phase change numbers of melting and vapourisation are defined as
(12)$$\;{\mathrm{Ph}}_{s/l}=\frac{L_{f}}{c_{p}\left(T_{v}-T_{0}\right)},\;\;{\mathrm{Ph}}_{l/v}=\frac{L_{v}}{c_{p}\left(T_{v}-T_{0}\right)}.$$
Employing the dimensionless variables in Equations (Equation 10) and (Equation 11), the governing equations (Equation 23)–(Equation 26) are rewritten in dimensionless form as
(13)$$\begin{array}{cc} \frac{\;dh^{*}}{\;dt^{*}} & =\Delta T^{*}+\frac{\left(1-R\right)\;\mathrm{La}}{\sigma^{*}\sqrt{2\pi }}\left\{2\cos^{2}\left[\frac{\pi x^{*}}{\Lambda^{*}}\right]\right\}\exp\left(-\frac{{\left(t^{*}-t_{p}^{*}\right)}^{2}}{2\sigma^{*\;2}}+\alpha^{*}\left(z^{*}-z_{\mathrm{surf}}^{*}\right)\right), \end{array}$$
(14)$$\begin{array}{cc} \nabla \cdot v^{*} & =0, \end{array}$$
(15)$$\begin{array}{cc} \frac{\;dv^{*}}{\;dt^{*}} & =-\nabla p_{\mathrm{dyn}}^{*}+\;\mathrm{Pr}\Delta v^{*}+\;\mathrm{Pr}\;\mathrm{Ra}\frac{T^{*}-T_{l}^{*}}{1-T_{l}^{*}}e_{z}, \end{array}$$
(16)$$\begin{array}{cc} \frac{\;dx^{*}}{\;dt^{*}} & =v^{*} \end{array}$$
with the dimensionless standard deviation $\sigma^{*}=\sigma /\left(4\tau_{p}\right)$ of the laser pulse, periodicity $\Lambda^{*}=\Lambda /L$, and absorption coefficient $\alpha^{*}=\alpha L$ used in Equation (Equation 13). Furthermore, the dimensionless form of the Marangoni boundary condition given in Equation (Equation 29) is obtained as [48]
(17)$$\frac{\partial v_{x}^{*}}{\partial z^{*}}=-\frac{\;\mathrm{Ma}}{1-T_{l}^{*}}\frac{\partial T^{*}}{\partial x^{*}}.$$

In addition to the aforementioned phase change numbers, $\;{\mathrm{Ph}}_{s/l}$ and $\;{\mathrm{Ph}}_{l/v}$ given in (Equation 12), and the Fourier number Fo corresponding to the considered physical time, the dimensionless numbers arising in the dimensionless Equations (Equation 13)–(Equation 17) are the laser number La
(18)$$\;\mathrm{La}=\frac{2\Phi_{0}\alpha }{\rho_{0}c_{p}\left(T_{v}-T_{0}\right)},\;\;\mathrm{Fo}=\frac{t_{\mathrm{end}}}{4\tau_{p}},$$
the Prandtl number Pr, the Rayleigh number Ra, and the Marangoni number Ma defined by
(19)$$\;\mathrm{Pr}=\frac{\nu }{a},\;\;\mathrm{Ra}=8\tau_{p}\frac{\beta \left(T_{v}-T_{l}\right)g\sqrt{a\tau_{p}}}{\nu },\;\;\mathrm{Ma}=-\frac{\;d\gamma }{\;dT}\frac{T_{v}-T_{l}}{\rho_{0}\nu }2\sqrt{\frac{\tau_{p}}{a}}.$$
At first, the Marangoni number Ma in the boundary condition (Equation 17) may be evaluated using the aforementioned temperature coefficient ${\left.\;d\gamma /\;dT\right|}_{\mathrm{pure}\mathrm{metal}}$ for liquid steel in the absence of the surfactant sulphur. In the presence of sulphur, the resulting Marangoni number Ma multiplied by the dimensionless temperature coefficient of surface tension given by (Equation 20), which is a function of temperature, is considered in the Marangoni boundary condition (Equation 17).
(20)$${\left(\frac{\;d\gamma }{\;dT}\right)}^{*}=\frac{\;d\gamma }{\;dT}{/\frac{\;d\gamma }{\;dT}|}_{\begin{array}{c} \mathrm{pure}\\ \mathrm{metal} \end{array}}=1-\frac{R\Gamma_{s}}{{\left.\frac{\;d\gamma }{\;dT}\right|}_{\begin{array}{c} \mathrm{pure}\\ \mathrm{metal} \end{array}}}\ln\left(1+Ka_{S}\right)-\frac{Ka_{S}}{1+Ka_{S}}\frac{\Gamma_{s}\Delta H_{0}/T}{{\left.\frac{\;d\gamma }{\;dT}\right|}_{\begin{array}{c} \mathrm{pure}\\ \mathrm{metal} \end{array}}}$$

### 2.5. Numerical Simulation

Simulation of the single pulse DLIP process considered material in the plane y=0 is subject to the central period of the interference pattern comprising the fluence maximum. According to the approach presented in [44], a two-dimensional section of 7.5 µm × 4.75 µm size in the *x*-*z*-plane was discretised using particles for the numerical solution by means of the smoothed particle hydrodynamics (SPH) method. The discretisation starts with coarse particles of 1 µm diameter at the bottom of the computational domain and a consecutive reduction of the particle diameter towards the surface according to a geometric sequence with common ratio 7/9; see Figure 4a. Near the surface, an initially equidistant array of fine particles of ${\left(7/9\right)}^{21}$ µm ≈ 5.1 nm diameter is arranged for the melt pool simulation; see Figure 4b. This minimum particle size is specified to ensure a complete insertion of the energy provided by the laser irradiation in accordance with an earlier investigation [49]. In the present discretisation, the equidistant arrangement of fine particles represents a surface layer of 270 nm depth. In addition, three rows and columns of dummy particles are situated beyond the domain boundaries to complete the kernel function support of nearby interior particles and to facilitate the application of boundary conditions.

The numerical investigation models the two-beam interference patterning process and setup described in Section 2.2 at moderate fluences. In detail, the process parameters considered for the numerical simulation of DLIP are presented in Table 1.

Note that the periodicity Λ is defined by the wavelength λ and the intersection angle θ; see Equation (Equation 1). In addition, two moderate laser fluences $\Phi_{\mathrm{av},\mathrm{spot}}=$ 0.300 J/cm^2^ and 0.375 J/cm^2^ were used in the numerical simulations. Considering an interference spot radius $r_{\mathrm{spot}}=$ 80 µm and a Gaussian beam radius $r_{0}=$ 100 µm, the average fluence in the central period of the interference pattern results from Equation (Equation 7) as $2\Phi_{0}=$ 0.532 J/cm^2^ and 0.665 J/cm^2^, respectively. Furthermore, simulations were performed for sulphur contents of 30 ppm, 100 ppm, 300 ppm, and 1500 ppm.

The considered material properties of the AISI 304 stainless steel substrates are compiled in Table 2. The temperature coefficient of surface tension dγ/dT given in the table represents the constant negative value for stainless steel in the absence of a surfactant, which is replaced by the model introduced in Section 2.4 in the presence of sulphur. Concerning the values of the density $\rho_{0}$, specific heat $c_{p}$, thermal conductivity κ, and thermal diffusivity *a* given in Table 2, the respective temperature-dependent values, if available, are averaged over the interval from the room temperature to the vapourisation point. The thermal diffusion length is then calculated employing the determined thermal diffusivity and presented in Table 3 along with the dimensionless quantities resulting from the material properties in Table 2.

## 3. Results and Discussion

This section reports and discusses the results of the DLIP experiments and the numerical simulations. In Section 3.1, the microstructures observed after laser processing, depending on the sulphur content of the substrate and the laser fluence, are presented. Furthermore, Section 3.2 provides the results of the simulations performed to comprehend the melt pool convection and the structure formation during the DLIP process.

### 3.1. Experimental Results

In a first set of experiments, the polished steel samples with different sulphur content were irradiated with DLIP line-like patterns by systematically changing the laser fluence. The DLIP spots showed differences in surface morphology for varying sulphur content. Figure 5 shows surface profiles of the steel samples with varied sulphur contents (30 ppm S, 100 ppm S, and 300 ppm S) treated at different laser fluences ($\Phi_{\mathrm{av},\mathrm{spot}}$ = 0.46 J/cm^2^, 0.63 J/cm^2^, 0.82 J/cm^2^, and 0.99 J/cm^2^). The grey highlighted area represents the region of maximum laser intensity (interference maximum) during structuring. For instance, the steel with 30 ppm sulphur treated with a laser fluence of $\Phi_{\mathrm{av},\mathrm{spot}}$ = 0.46 J/cm^2^ exhibits a line-like pattern with a central peak (∼40 nm height) surrounded by two valleys (∼15 nm depth) below the initial surface level. Increasing the laser fluence to $\Phi_{\mathrm{av},\mathrm{spot}}$ = 0.82 J/cm^2^ leads to a transformation stage, where the central single peak splits into two peaks with a structure height of approximately 50 nm, forming a well-defined valley between them. By further increasing the laser fluence, both the height of the peaks and the depth of the valley increases, determined by a significant flow of the molten material during the DLIP process; see Figure 5 for 30 ppm S content and $\Phi_{\mathrm{av},\mathrm{spot}}$ = 0.99 J/cm^2^.

The line-like pattern on the steel with 100 ppm S reveals a topography similar to the processed one of 30 ppm S steel when irradiated at a low fluence level of $\Phi_{\mathrm{av},\mathrm{spot}}$ = 0.46 J/cm^2^, with similar peak heights and valley depths; see Figure 5. However, the central peak, with a maximum structure height of 50 nm as well, starts to split only at laser fluences above $\Phi_{\mathrm{av},\mathrm{spot}}$ = 0.63 J/cm^2^. By fluence increase to $\Phi_{\mathrm{av},\mathrm{spot}}$ = 0.99 J/cm^2^, both peak height and valley depth constantly increase further and reach a similar magnitude as that for the geometries on 30 ppm S steels irradiated with $\Phi_{\mathrm{av},\mathrm{spot}}$ = 0.99 J/cm^2^.

In contrast, for the steel samples with the highest concentration of sulphur (300 ppm and 1500–3000 ppm), a different behaviour was observed at low fluences. Here, two separate peaks with peak heights of up to 50 nm instead of a centre peak are found also at low laser fluences; see Figure 5 for 300 ppm S content and $\Phi_{\mathrm{av},\mathrm{spot}}$ = 0.46 J/cm^2^. As the laser fluence is increased, the split peaks become larger, reaching up to 50 nm absolute structure height at $\Phi_{\mathrm{av},\mathrm{spot}}$ = 0.82 J/cm^2^. At $\Phi_{\mathrm{av},\mathrm{spot}}$ = 0.99 J/cm^2^, the height of the split peaks is further increased, similar to the 30 ppm and 100 ppm samples.

The surface microstructures generated on steel samples with 30 ppm, 100 ppm, and 300 ppm S using laser fluences in the range 0.46 J/cm^2^–0.99 J/cm^2^ are further illustrated by the scanning electron micrographs shown in Figure 6. The small graphs inserted into the SEM images in Figure 6 indicate the areas of maximum laser intensity on each surface.

In agreement with the surface profiles presented in Figure 5, the micrographs of steels with 30 ppm S and 100 ppm S contents irradiated with a laser fluence of 0.46 J/cm^2^ in Figure 6 depict structures with a single central peak. With an increase in laser fluence, the structure width grows, where a smaller central peak and two neighbouring subpeaks evolve instead of the single peak structure. On the other hand, microstructures fabricated using the highest laser fluence of 0.99 J/cm^2^ display a split peak formation, induced by the dominant recoil pressure during the patterning process. In contrast, the SEM images of structures made on steel substrates with 300 ppm S in Figure 6 show a split peak formation for all laser fluences applied, where the structure width increases with the laser fluence up to 0.99 J/cm^2^.

The differences in peak formation can also be identified in the three-dimensional images presented in Figure 7. Small, single peaks of the structures obtained on 30 ppm S steel at a laser fluence of $\Phi_{\mathrm{av},\mathrm{spot}}$ = 0.46 J/cm^2^ are shown in Figure 7a. The topography shown in Figure 7b on steel with 300 ppm S after DLIP treatment at $\Phi_{\mathrm{av},\mathrm{spot}}$ = 0.82 J/cm^2^ is characterised by split peaks and an enclosed valley below the initial surface level. The structure heights are decreasing from the DLIP spot centre towards the spot edge, following the Gaussian intensity distribution of the laser beam. Even at the spot edge, no structures in transformation stage or single peaks are visible in Figure 7b.

An explanation of the distinct structure formation and the different peak heights at similar laser fluence is attempted on the basis of the surface tension gradient depending on the sulphur content of the melted steel [47]. As evident from Equation (Equation 9) and Figure 3b, both the surface temperature and the sulphur activity affect the temperature coefficient of surface tension, which determines the direction of thermocapillary flow and the melt pool geometry [38]. For 30 ppm S, the surface tension temperature coefficient is positive only in a small temperature range of ∼300 K from the liquidus point $T_{l}$. At moderate fluences, the positive temperature derivative of surface tension induces an inward flow at the surface and the formation of a single peak structure with adjacent depressions, as shown in Figure 1b, e.g., for 30 ppm S and $\Phi_{\mathrm{av},\mathrm{spot}}$ = 0.46 J/cm^2^ in Figure 5. For an augmented sulphur content of 100 ppm, the magnitude and temperature range of the positive temperature coefficient of surface tension is enlarged, resulting in a stronger inward convection and more marked single peak structures at moderate fluences, as presented in Figure 5 for $\Phi_{\mathrm{av},\mathrm{spot}}$ = 0.46 J/cm^2^ and 0.63 J/cm^2^.

With an increase in laser fluence, the higher surface temperatures are associated with a negative temperature coefficient of surface tension of the steel melt at low sulphur concentrations. Accordingly, the inward flow from the melt pool edges is temporarily suppressed by an outward flow from the centre of the melt pool surface at higher surface temperatures, which causes the evolution of outer peaks near the edges of the melt pool; see Figure 1a. In addition, small central peaks may form due to inward convection in the late stages of the melt pool presence at lower melt surface temperatures; see the microstructures for 30 ppm S at $\Phi_{\mathrm{av},\mathrm{spot}}$ = 0.63 J/cm^2^ and 0.82 J/cm^2^ and for 100 ppm S at $\Phi_{\mathrm{av},\mathrm{spot}}$ = 0.82 J/cm^2^ in Figure 5. On the contrary, it is evident from the split peak structures observed in Figure 5 for steel with 300 ppm S at fluences up to $\Phi_{\mathrm{av},\mathrm{spot}}$ = 0.82 J/cm^2^ that the conception of an even stronger inward convection at higher sulphur contents is not confirmed by the experiments. This deviation suggests that, in the presence of higher impurity levels, the expression for the temperature-dependent surface tension in Equation (Equation 8) is no longer appropriate or that effects other than thermocapillary convection are responsible for the microstructure evolution. For a high fluence of $\Phi_{\mathrm{av},\mathrm{spot}}$ = 0.99 J/cm^2^, split peak structures of similar height are produced on all steel samples, independent of the sulphur content; see Figure 5. This observation may be attributed to the dominance of the recoil pressure induced by vapourisation of molten metal at high laser intensity, which causes a lateral ejection of melt during the patterning process [56].

It is important to mention the different behaviour observed at the fluences below $\Phi_{\mathrm{av},\mathrm{spot}}$ = 0.46 J/cm^2^ or at the edges of the DLIP spots. In these cases, single peaks are observed without the surrounding valleys, as compared to the centre peaks of the 30 ppm and 100 ppm S samples treated with $\Phi_{\mathrm{av},\mathrm{spot}}$ = 0.46 J/cm^2^. In other words, this structure has been produced without the flow of molten metal and thus can only be explained by the transformation of the crystalline structure of the used steel. When an austenitic steel is heated, depending on the temperature reached and the cooling rate, the structure can transform to martensite, which has a lower density (austenite: 7.9 g/cm^3^; martensite: 6.5 g/cm^3^ [57]). Taking into consideration that the measured height of the peak was 15 nm, it is estimated that the austenitic steel was affected up to a depth of 69 nm.

The dependence of structure heights and peak splitting on sulphur content and laser fluence is shown in Figure 8. The change in structure height between centre peak and initial surface level is depicted in Figure 8a. Fluences below $\Phi_{\mathrm{av},\mathrm{spot}}$ = 0.46 J/cm^2^ produce slight surface elevations related to phase changes austenite–martensite in the steel. A fluence increase from $\Phi_{\mathrm{av},\mathrm{spot}}$ = 0.46 J/cm^2^ to $\Phi_{\mathrm{av},\mathrm{spot}}$ = 0.9 J/cm^2^ results in a gain in structure heights, up to 49 nm for single peaks with surrounding valleys for <100 ppm S steels. A further rise in fluence only results in a decrease in structure height due to a transformation stage from single peak to split peaks. Structures on steel with higher sulphur content (>100 ppm S) show negative structure heights, that is, valleys at the centre, for fluences above $\Phi_{\mathrm{av},\mathrm{spot}}$ = 0.46 J/cm^2^ due to the split peak formation. With increasing fluence to $\Phi_{\mathrm{av},\mathrm{spot}}$ = 0.99 J/cm^2^, the valley enclosed by the split peaks becomes slightly deeper and increases further for fluences above $\Phi_{\mathrm{av},\mathrm{spot}}$ = 0.99 J/cm^2^. Maximum peak heights of 150 nm are found for $\Phi_{\mathrm{av},\mathrm{spot}}$ = 1.1 J/cm^2^ on all steel samples, irrespective of the sulphur content.

The topic of peak splitting is also addressed in Figure 8b. Here, the distance between the split peaks is analysed. No peak splitting is detected for fluences below $\Phi_{\mathrm{av},\mathrm{spot}}$ = 0.46 J/cm^2^. Starting from $\Phi_{\mathrm{av},\mathrm{spot}}$ = 0.46 J/cm^2^, a separation distance (half distance between both peaks) increasing with the laser fluence up to a maximum of 2 µm is measured for high sulphur content steels.

### 3.2. Simulation Results

Simulations of single pulse DLIP were performed for stainless steel substrates with different sulphur contents. As mentioned in Section 2.5, the numerical simulations considered the process parameters and material properties given in Table 1 and Table 2, respectively, as well as the resulting diffusion length and dimensionless quantities in Table 3. The dimensionless form (Equation 20) of the surface tension temperature coefficient in Equation (Equation 9) was incorporated into the Marangoni boundary condition in Equation (Equation 17). Furthermore, simulations were carried out for the central period of the interference pattern with an average fluence of either $2\Phi_{0}=$ 0.532 J/cm^2^ or $2\Phi_{0}=$ 0.665 J/cm^2^, corresponding to a laser number of La=18.0241 or La=22.5301.

The evolution of the maximum surface temperature predicted in the simulation for the moderate laser fluence $2\Phi_{0}=$ 0.532 J/cm^2^ is presented in Figure 9 along with the temporal variation of the laser pulse intensity. The maximum surface temperature trend in Figure 9 is in line with earlier simulation results in [44], although $T_{\max}∼2570$ K is lower here, for DLIP of stainless steel at a laser wavelength of λ=355 nm and a lower fluence of $2\Phi_{0}=$ 0.4 J/cm^2^, which is attributed to the higher reflectivity at the present wavelength. It is evident from Figure 9 that the substrate surface is heated up to a temperature significantly above the liquidus point at the interference maximum due to the action of the laser pulse. Accordingly, a melt layer with the dimensions computed by the SPH model and presented in Figure 10 develops near the interference maximum after the onset of the laser pulse. Concerning the trends in Figure 10, the discrete values of the melt pool dimensions are employed only once at a central point in time to avoid a stair-step appearance of the graphs. The melt pool depth and the duration of the melt presence are compatible with the aforementioned numerical results in [44], whereas the melt pool width in Figure 10 exceeds the earlier calculations owing to the larger periodicity Λ of the interference pattern considered here.

Furthermore, simulations were performed for an elevated laser fluence 2$\Phi_{0}=0.665$ J/cm^2^ to investigate the influence of the laser fluence on the material behaviour, in particular the nonlinear effect on the melt pool dimensions and the velocity field, and the structure formation reported in Section 3.1. Consequently, the evolution of the maximum surface temperature computed by the SPH model is also shown in Figure 9. The temporal maximum of the surface temperature $T_{\max}∼3150$ K at the interference maximum is below the vapourisation point, unlike the simulation results in [44] for the comparable fluence 2$\Phi_{0}=0.5$ J/cm^2^ and the laser wavelength λ=355 nm with a lower reflectivity of the stainless steel substrate. These differences from the earlier results are attributed to the slightly larger width of the laser pulse considered in the present investigation. The computed dimensions of the melt layer developing after the beginning of the laser pulse are shown in Figure 10 for the elevated fluence 2$\Phi_{0}=0.665$ J/cm^2^ as well. In accordance with the numerical results in [44], the maximum melt pool dimensions are roughly 20% wider and even 50% deeper than the calculations for the moderate fluence, i.e., 2$\Phi_{0}=0.532$ J/cm^2^ here. Again, the duration of the melt presence and the determined melt pool depth conform with the previous results in [44] unlike the increased melt pool width due to the larger periodicity Λ employed.

As set out in the Introduction, the nonuniform surface temperature of the melt layer gives rise to surface tension gradients, entailing thermocapillary convection in the direction of high surface tension. Considering the presence of the surfactant sulphur in liquid steel, the temperature coefficient of surface tension is positive near the melting point, i.e., there is inward convection from the melt pool edges towards regions of higher surface tension and temperature. The computed velocity magnitude of this inward flow at the melt pool surface is indicated by the solid lines in Figure 11a for different sulphur contents and the moderate laser fluence $2\Phi_{0}=0.532$ J/cm^2^. If the surface of the melt pool is heated to higher temperatures, the temperature coefficient of surface tension is negative in the proximity of the interference maximum and an additional outward flow from the centre of the melt pool surface towards regions of maximum surface tension develops. The predicted velocity magnitude of this outward convection is indicated by the dashed lines in Figure 11a for different sulphur contents.

It is observed from Figure 11a that the magnitude of the outward flow is particularly prominent for a low sulphur concentration of 30 ppm and decreases with an increase in sulphur content. On the contrary, the inward flow becomes more pronounced for a higher surfactant concentration, as can be identified from Figure 11a. For 1500 ppm sulphur, the highest sulphur content investigated, the thermocapillary convection is completely inward at the melt pool surface for the moderate laser fluence, as the maximum surface temperature does not exceed the point of maximum surface tension; see also Figure 3 and Figure 9. For a clearer idea of the substrate behaviour in the interaction zone, Figure 12a presents the predicted temperature and velocity fields in a section of the computational domain comprising the melt pool half width for different sulphur contents at several times. In addition to the velocity magnitudes shown in Figure 11a for the moderate fluence 2$\Phi_{0}=0.532$ J/cm^2^, Figure 12a illustrates the strong outward convection for a low sulphur content of 30 ppm, which gradually decreases and is dominated by the inward convection for higher sulphur concentrations of 100 ppm and 300 ppm. Compared with the results in [44], lower horizontal velocity magnitudes are obtained in this work, which is attributed to the generally smaller magnitude of the surface tension temperature coefficient in the presence of a surfactant and the lower surface temperature gradients due to the larger periodicity Λ and longer pulse duration $\tau_{p}$.

The magnitudes of the horizontal velocity extrema at the melt pool surface predicted for the elevated laser fluence $2\Phi_{0}=0.665$ J/cm^2^ are presented in Figure 11b. It is noted in Figure 11b that the velocity magnitude of the inward flow from the melt pool edges is consistently higher than the results in Figure 11a for the moderate laser fluence. As the surface temperature largely exceeds the point of maximum surface tension in the central region due to the elevated fluence $2\Phi_{0}=0.665$ J/cm^2^, the outward flow from the centre of the melt pool surface is significantly enhanced; see Figure 11b. Considering a low sulphur content of 30 ppm, Figure 11b shows that the outward flow clearly dominates the melt pool convection. For a concentration of 100 ppm sulphur in liquid steel, outward and inward convection both contribute considerably to the melt pool flow pattern. On the contrary, the graphs in Figure 11b suggest that the inward flow still dominates the melt pool convection for a higher sulphur content of 300 ppm in spite of the augmented outward flow. The foregoing statements are further illustrated in Figure 12b, which depicts the temperature and velocity fields computed for DLIP of stainless steel employing the elevated fluence in a section comprising the melt pool half width. In particular, a comparison of the results for 100 ppm sulphur presented in the central columns of Figure 12a,b reveals that a slight increase in the laser fluence changes the melt pool convection from a predominant inward flow to competing outward and inward flow. This changed melt flow character may explain the transition from a single peak microstructure to split peaks in Figure 5 when augmenting the laser fluence.

## 4. Conclusions

Direct laser interference patterning was employed on steel substrates with varied sulphur content to study the influence of melt dynamics on the produced surface topographies. For the structuring process, a laser source with a wavelength of 1053 nm and a pulse duration of 12 ns was selected to permit a photothermal interaction at the steel surface and thus the melt pool formation. At laser fluences below $\Phi_{\mathrm{av},\mathrm{spot}}$ = 0.46 J/cm^2^, low peak heights were produced independently of the sulphur content. An explanation can be found in the phase change in the steel microstructure from austenite to martensite, which involves a volume expansion due to the lower density of martensite. A slight enhancement of the laser fluence resulted in an increase in height of the single peak structures formed on low sulphur steels, which can be related to the inverse Marangoni convection in the melt pool. With a further increase in laser fluence, the structures on low sulphur steels change from single peak to split peaks. In contrast, for steels with high sulphur content, structures with split peaks were produced also at low laser fluences. Structures generated employing laser fluences above $\Phi_{\mathrm{av},\mathrm{spot}}$ = 0.90 J/cm^2^ exhibited split peaks, irrespective of the sulphur content, due to the dominance of the recoil pressure during the DLIP process. The difference in the surface topography observed as a function of the laser parameters and sulphur content can be used in the future to create pattern geometries different from the yet achievable with two-beam or three-beam DLIP and thus to produce surfaces with novel functions.

As the process is not accessible to measurement, numerical simulation is used to gain insight into the mechanisms effective during nanosecond pulsed DLIP of stainless steel with sulphur content. For this purpose, an SPH model for heat transfer and fluid flow during the patterning process is extended to take the nonlinear temperature dependence of surface tension in the presence of a surfactant into account. Considering the process parameters and physical properties of the substrates employed in the experiments, process simulations of DLIP on stainless steel with different sulphur content were performed. The computational results provide detailed information on the temperature field, the melt pool size, and flow pattern in the interaction zone. In particular, the influence of sulphur concentration on the thermocapillary convection and the effect of laser fluence on the melt pool dimensions and flow pattern were investigated numerically. Consequently, the simulations support the explanation of the microstructure formation observed in the experiments. For instance, the evolution of both single peak structures at moderate fluence and split peak structures at elevated fluence found after DLIP on stainless steel with 100 ppm sulphur is confirmed by the dominating inward flow and the competing outward flow, respectively, in the SPH simulations.

The simulation results hint at the relevance of the melt pool flow in the microstructure evolution due to nanosecond pulsed DLIP of metals at moderate fluences, although the melt displacement and the structure formation cannot be reproduced at present. Prospective numerical investigations of nanosecond pulsed DLIP on metal substrates will include the modelling of the surface deformation and the vapourisation recoil pressure at higher fluences.

## Figures and Tables

**Figure 1 nanomaterials-11-00855-f001:**
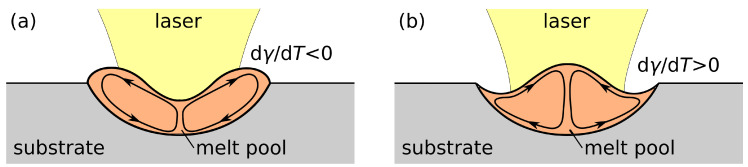
Surface evolution during laser processing due to temperature gradients and thermocapillary flow for (**a**) negative and (**b**) positive temperature coefficients of surface tension dγ/dT, outlines adopted from [32,43].

**Figure 2 nanomaterials-11-00855-f002:**
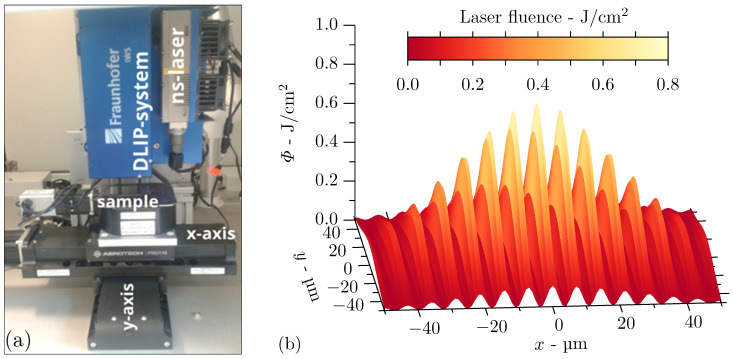
Setup employed for direct laser interference patterning (DLIP) experiments on steel substrates: (**a**) nanosecond pulsed infrared laser, optical head for DLIP on a *z*-stage, and sample mounted on an *x*-*y* positioning stage and (**b**) laser fluence distribution of interference pattern due to two coherent partial beams with a Gaussian intensity profile.

**Figure 3 nanomaterials-11-00855-f003:**
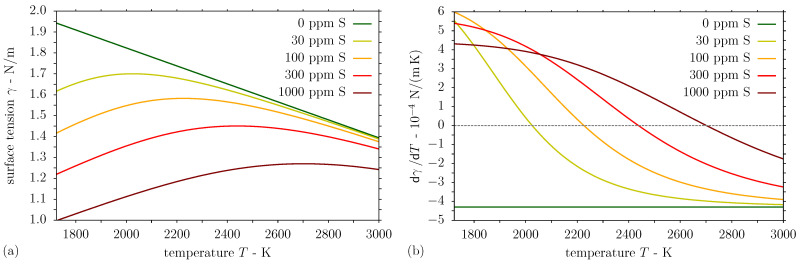
Temperature-dependent (**a**) surface tension and (**b**) its temperature derivative for liquid steel with different sulphur contents.

**Figure 4 nanomaterials-11-00855-f004:**
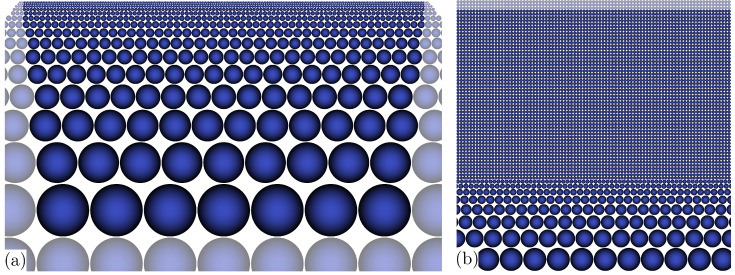
Particle discretisation of computational domain, dummy particles less opaque, details: (**a**) 7.5 µm × 4.4 µm, coarse particles start at the bottom, (**b**) 440 nm × 440 nm, near-surface equidistant initial array and coarsening.

**Figure 5 nanomaterials-11-00855-f005:**
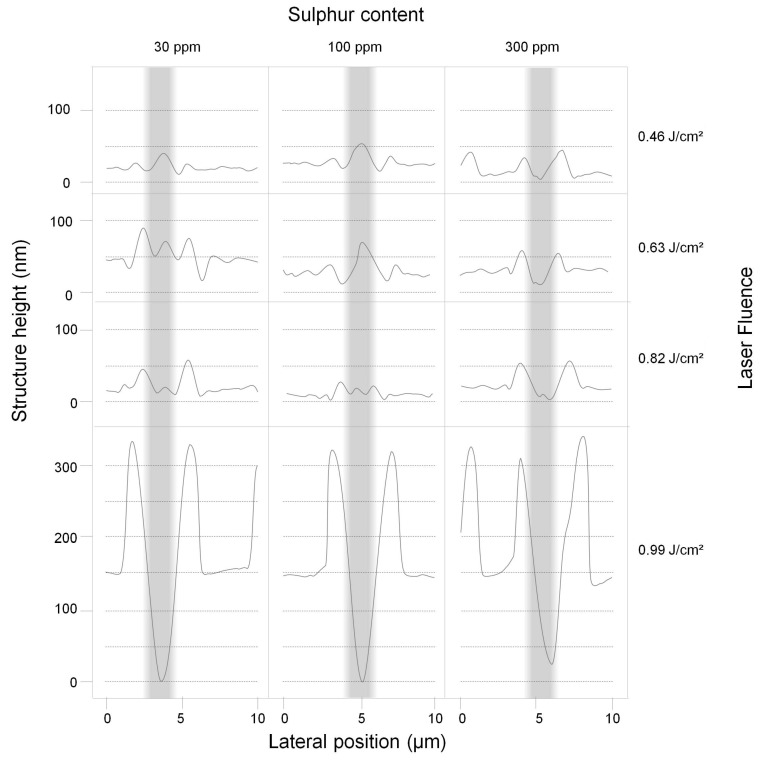
Overview of peak transformation of DLIP structures made on steels with varying sulphur contents. The columns show profiles of produced patterns in steels with sulphur contents of 30 ppm, 100 ppm, and 300 ppm at different laser fluences ($\Phi_{\mathrm{av},\mathrm{spot}}$ = 0.46 J/cm^2^, 0.63 J/cm^2^, 0.82 J/cm^2^, and 0.99 J/cm^2^). The grey highlighted areas represent the regions of maximum laser intensity during patterning.

**Figure 6 nanomaterials-11-00855-f006:**
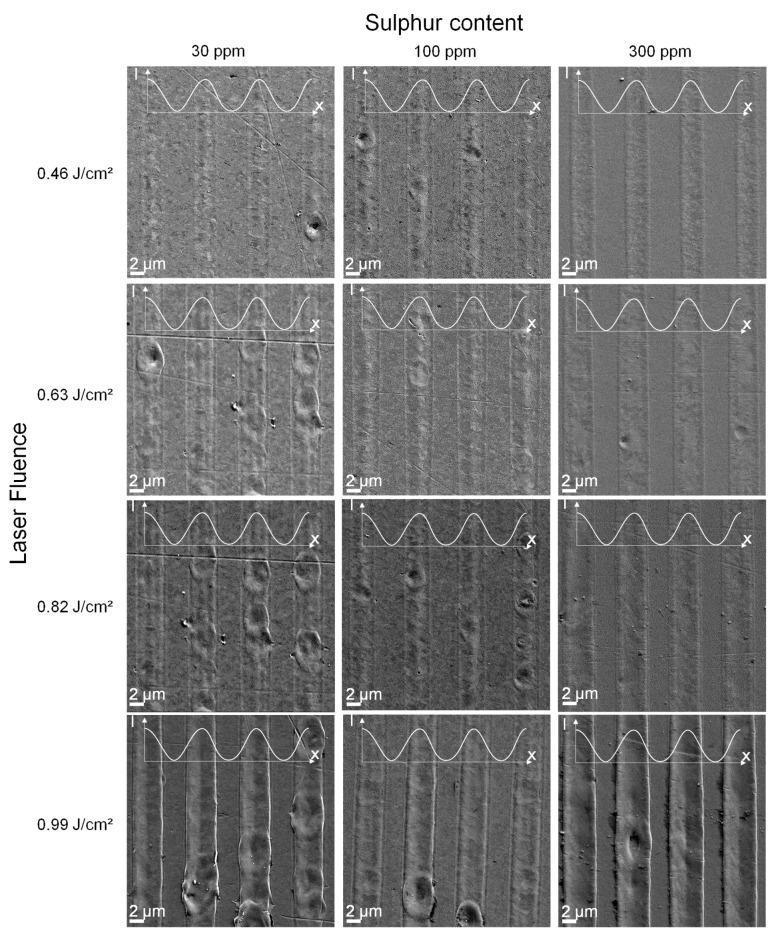
Scanning electron micrographs of surfaces after DLIP (λ = 1053 nm, Λ = 7.2 µm, and $\tau_{p}$ = 12 ns) on steel substrates with distinct sulphur contents using a single pulse and different laser fluences. Small graphs indicate areas of maximum laser intensity. The brightness and contrast of the SEM images were enhanced for better visualisation.

**Figure 7 nanomaterials-11-00855-f007:**
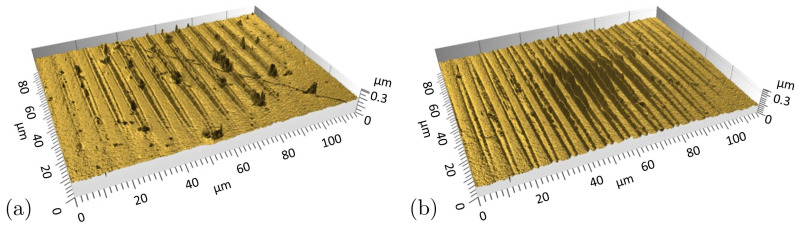
Three-dimensional confocal microscope images of DLIP laser spots on steel samples with different sulphur contents. (**a**) Single peak structures generated on steel with 30 ppm S content and $\Phi_{\mathrm{av},\mathrm{spot}}$ = 0.46 J/cm^2^, (**b**) split peak structures fabricated on steel with 300 ppm S content and $\Phi_{\mathrm{av},\mathrm{spot}}$ = 0.82 J/cm^2^.

**Figure 8 nanomaterials-11-00855-f008:**
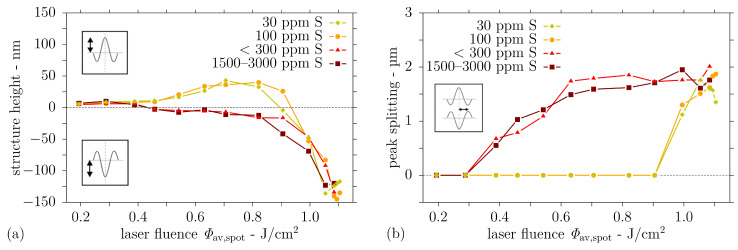
Surface topography observed after single pulse DLIP (λ = 1053 nm, Λ = 7.2 µm, $\tau_{p}$ = 12 ns) on steel with different sulphur contents as a function of fluence: (**a**) structure height and (**b**) distance of peak(s) from centre.

**Figure 9 nanomaterials-11-00855-f009:**
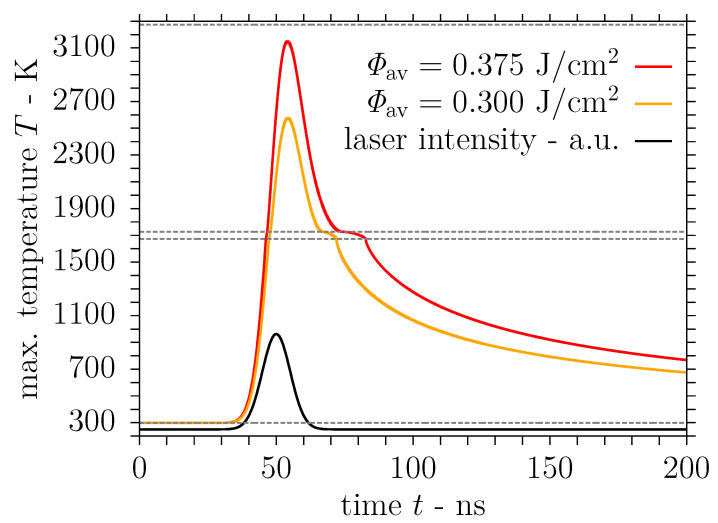
Maximum surface temperature during single pulse DLIP of stainless steel with Gaussian beams using the process parameters λ = 1053 nm, Λ = 7.2 µm, $\tau_{p}$ = 12 ns and $2\Phi_{0}$ = 0.532 J/cm^2^ or $2\Phi_{0}$ = 0.665 J/cm^2^.

**Figure 10 nanomaterials-11-00855-f010:**
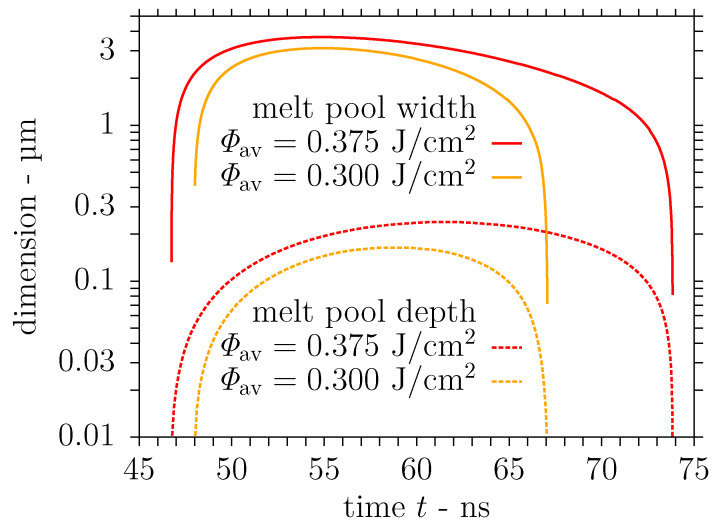
Transient melt pool dimensions predicted by the smoothed particle hydrodynamics (SPH) model of DLIP (λ = 1053 nm, Λ = 7.2 µm, $\tau_{p}$ = 12 ns) on stainless steel using a laser fluence of either $2\Phi_{0}$ = 0.532 J/cm^2^ or $2\Phi_{0}$ = 0.665 J/cm^2^.

**Figure 11 nanomaterials-11-00855-f011:**
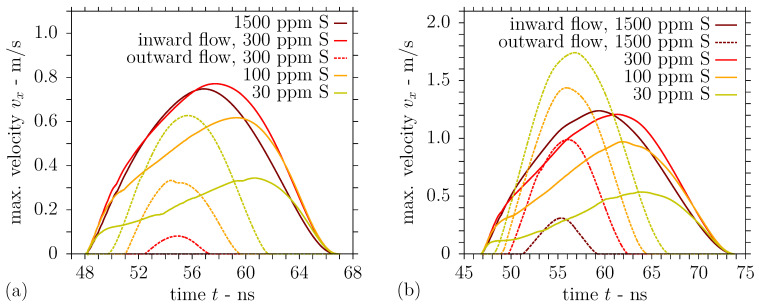
Horizontal velocity magnitudes at melt pool surface during DLIP with λ = 1053 nm, Λ = 7.2 µm, and $\tau_{p}$ = 12 ns using a fluence of (**a**) $2\Phi_{0}$ = 0.532 J/cm^2^ or (**b**) $2\Phi_{0}$ = 0.665 J/cm^2^ on steel with sulphur content.

**Figure 12 nanomaterials-11-00855-f012:**
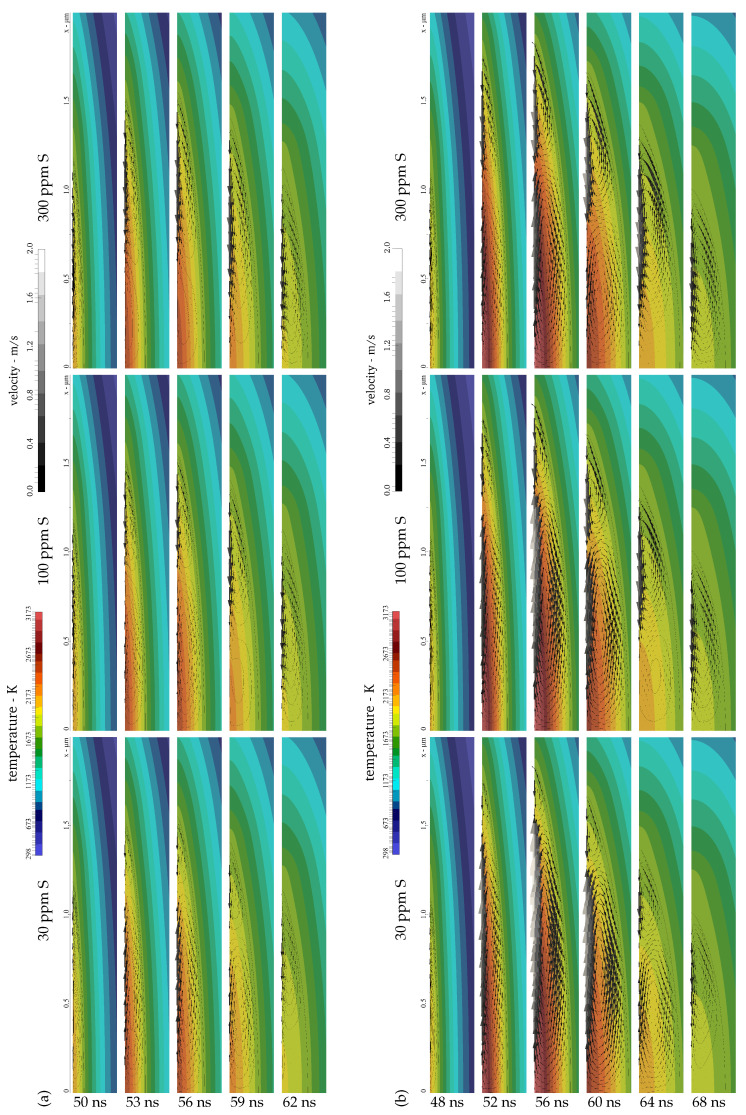
Simulation of melt pool flow during DLIP (λ = 1053 nm, Λ = 7.2 µm, and $\tau_{p}$ = 12 ns) of steel with (l) 30 ppm, (c) 100 ppm, and (r) 300 ppm sulphur at a laser fluence of (**a**) $2\Phi_{0}$ = 0.532 J/cm^2^ and (**b**) $2\Phi_{0}$ = 0.665 J/cm^2^ with detail of 2 µm × 250 nm including isotherms, streamlines, and velocity vectors in melt pool half width at times (**a**) *t* = 50 ns, 53 ns, 56 ns, 59 ns, and 62 ns, and (**b**) *t* = 48 ns, 52 ns, 56 ns, 60 ns, 64 ns, and 68 ns.

**Table 1 nanomaterials-11-00855-t001:** Process parameters of single pulse DLIP on stainless steel with sulphur content.

Process Parameter	Symbol	Value
wavelength	λ	1053 nm
intersection angle between beams	θ	0.1464 rad
periodicity of interference pattern	Λ	7.2 µm
average fluence in interference spot	$\Phi_{\mathrm{av},\mathrm{spot}}$	0.300 J/cm^2^
fluence of interference pattern	$2\Phi_{0}$	0.532 J/cm^2^
pulse duration (FWHM)	$\tau_{p}$	12 ns
pulse time	$t_{p}$	50 ns
simulation duration	$t_{\mathrm{end}}$	200 ns
initial substrate temperature	$T_{0}$	298.15 K
gravitational acceleration	*g*	9.81 m/s^2^
Fourier number	Fo	$4.1\bar{6}$

**Table 2 nanomaterials-11-00855-t002:** Material properties of stainless steel substrates.

Material Property	Symbol	AISI 304 Steel	Unit	References
solidus temperature	$T_{s}$	1673	K	[50]
liquidus temperature	$T_{l}$	1727	K	[50]
vapourisation temperature	$T_{v}$	3273	K	[51]
density	$\rho_{0}$	7262	kg/m^3^	[50]
specific heat	$c_{p}$	704	J/(kg K)	[50,52]
thermal conductivity	κ	26.8	W/(m K)	[50]
thermal diffusivity	*a*	$5.24\times 10^{-6}$	m^2^/s	
enthalpy of fusion	$L_{f}$	251	kJ/kg	[50,51]
enthalpy of vapourisation	$L_{v}$	6500	kJ/kg	[51]
dynamic viscosity (at $T_{l}$)	η	$7.0\times 10^{-3}$	Pa s	[50]
kinematic viscosity (at $T_{l}$)	ν	$1.02\times 10^{-6}$	m^2^/s	
volumetric thermal expansion coefficient	β	$8.5\times 10^{-5}$	1/K	[53,54]
temperature coefficient of surface tension	$\;d\gamma /\;dT$	$-4.3\times 10^{-4}$	N/(m K)	[45]
absorption coefficient (at 1053 nm)	α	$5.15\times 10^{7}$	1/m	[55]
reflectivity (at 1053 nm)	*R*	0.646	1	[55]

**Table 3 nanomaterials-11-00855-t003:** Diffusion length and dimensionless numbers of DLIP model for stainless steel surfaces.

Quantity	Symbol	AISI 304 Steel
thermal diffusion length	*L*	501.5 nm
laser number	La	18.0241
solid–liquid phase change number	$\;{\mathrm{Ph}}_{s/l}$	0.119849
liquid–vapour phase change number	$\;{\mathrm{Ph}}_{l/v}$	3.10367
Prandtl number	Pr	0.1947
Rayleigh number	Ra	$3.04124\times 10^{-8}$
Marangoni number	Ma	9.08939

## Data Availability

The data presented in this article are available upon request from the authors.

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
