# Peer review of "Influence of Sulphur Content on Structuring Dynamics during Nanosecond Pulsed Direct Laser Interference Patterning"

_nanomaterials, 2021, doi:10.3390/nano11040855_

Round 1

Reviewer 1 Report

This is an interesting, well-written manuscript describing an excellent set of experiments on direct laser interference patterning.  I am happy to recommend publication in Nanomaterials.  The authors may want to add DLIP to their list of abbreviations/acronyms on page 19. 

Author Response

We would like to thank the Reviewer for her/his positive assessment of our manuscript and the appreciation of the DLIP experiments.

In accordance with her/his valuable advice, we have now included the acronym DLIP for "direct laser interference patterning" in the list of abbreviations on page 19 of the revised manuscript.

Reviewer 2 Report

This manuscript systematically and thoroughly investigated the influence of structural dynamics of DLIP on sulfur contents in steel and laser fluence. Both experiments and results are very clear, and the manuscript is well-written. The reviewer believes that this manuscript can be published in Nanomaterials as it is. The following is minor comments.

Minor comments

  1. It may be confusing to use energy density in Fig. 2(b), page 2,4, 5, 14, 15, 16. In general the unit of energy density is [J/cm^3], and the author uses [J/cm^2] for the unit of energy density, the same as the unit of laser fluence. The reviewer believes that it is better for the authors to use laser fluence instead of energy density.
  2. In Eq.(3)-(7), the authors defines the Gaussian beam radius as r_0. For DLIP, two laser beams are incident at off normal incidence, and the shape of Gaussian spot is most likely elliptical. Although the incident angle of laser beam used in this manuscript is small enough to assume that the shape of spot is circular, it would be great at least to mention their assumption in the manuscript.

Author Response

Dear Reviewer,

Please see the responses to the comments in the attachment.

Reviewer 3 Report

Influence of sulphur contents on texturing metallic steel surfaces using a nanosecond pulsed laser source and direct laser interference patterning (DLIP) technique is described in this manuscript. Steel surfaces with distinctly different active element sulphur contents affects the melt convection pattern and the pool shape during the process. The analysis highlights different topographic features that can be produced and associated melt dynamics, presented for varied sulphur concentrations. The results indicate single peak geometries formed on substrates with lower sulphur contents, while split peaks structures were observed in higher sulphur content steels (>300 ppm). The peak formation is explained using weld pool thermocapillary convection and further supported using  numerical simulation studies based on smoothed particle hydrodynamics model. The manuscript is well written with all the results presented methodically.

Author Response

We are grateful to the Reviewer for the assessment and approval of the manuscript and appreciate the favourable statements. At the same time, we would like to state precisely that split peak structures were also generated in the DLIP experiments using moderate laser fluences on the steel substrates with no more than 300 ppm sulphur content.

Reviewer 4 Report

Dear Authors,

I read your article with great interest even though it is quite huge. I found it very well-written. You prepared an interesting Introduction, well supported by the huge number of references. The experiments were very thoroughly described. The artwork was clear and informative. So, my overall opinion is the article fits all standards and requirements to be further processed and finally, published. Please find below some minor remarks and address them. After that, I recommend the article for publishing.

Sincerely,

Reviewer

  1. The number of keywords 9 seems to be quite big. I wonder if it is necessary?
  2. The first parts of the Introduction and the citations 1-5 and some others are rather uncommon for technical papers, however, I did not find it wrong.
  3. Equations 6, 7, 8, etc.: There are some dots, commas after the formulas. Why? I believe this is an editorial error?
  4. Line 200: It is worth mentioning there A9 is one of the formulas given in the Appendix. Then the next similar citations will be totally clear.
  5. What was the software to perform the simulation?

Author Response

Dear Reviewer,

Thank you for your recommending report. We appreciate the time you spent examining our manuscript and your positive comments on the introduction, the experiments and the artwork. Furthermore, we acknowledge your above remarks, which allowed us to revise the manuscript. Please find our responses to the remarks below.

Yours faithfully,
the Authors

  1. We confirm that the number of nine keywords in the submitted manuscript may be unnecessarily large and distracting. Correspondingly, we have removed the two less substantiated keywords "laser material processing" and "stainless steel", and slightly reordered the remaining seven keywords in the revised manuscript.
  2. With regard to the introduction, we are aware that the first paragraphs and some of the references cited therein could be considered as distantly related to the object of investigation. Nevertheless, we believe that these passages are beneficial as they put the manuscript into a greater context.
  3. According to our understanding, displayed equations are in general part of the text. More precisely, formulae in display mode are part of the preceding sentence and thus punctuated in the same way as if they were inline formulae. We are hopeful that the publisher is in a position to agree with our opinion on this typesetting question.
  4. We are thankful for this comment aiming at the clarity of presentation. In the revised manuscript, we have mentioned in line 201, following the reference to this boundary condition, that condition (A9) is given in Appendix A.
  5. For the simulations performed in the scope of this article, a numerical code developed by the second author (in C++) was extended to accommodate the modification of the Marangoni boundary condition (17) by the dimensionless temperature coefficient of surface tension in Eq. (20). The numerical technique implemented in this code is based on the SPH method as presented in our (more) fundamental paper in Ref. [44]. The simulation data were saved to files and subsequently visualised using either gnuplot for the transient data in Figs. 9, 10 and 11 or ParaView for the field data in Figs. 4 and 12.

Reviewer 5 Report

The paper “Influence of Sulphur Content on Structuring Dynamics during Nanosecond Pulsed Direct Laser Interference Patterning” provides a report on a systematic study of nanosecond laser structuring of Steel with Sulphur content to produce surface topographies and understand the physics behind the formation by performing appropriate simulations. Although the work is scientifically sound and is of importance in the field of laser welding and micromachining, it does not suit with the aim and scope of the Nanomaterials Journal. 

Author Response

We would like to thank the Reviewer for acknowledging our manuscript. With regard to the remark about the Aims & Scope of the journal Nanomaterials, we would like to point out the vertical amplitude of the periodic surface structures generated in our work due to the melt flow during nanosecond pulsed DLIP, depending on the sulphur content of the steel substrate and the laser fluence. Here, nanometric structure sizes ranging from 15 nm valley depth for substrates with low sulphur content up to 300 nm overall peak-to-valley height for high laser fluences are measured. Furthermore, we would like to mention the peak transformation from single peak to splitted peaks with up to 4 µm peak-to-peak distance.

According to our understanding, the structure heights in the lower nanometre size range fit into the scope of Nanomaterials. Incidentally, we have identified the article by Yang et al. "Periodic microstructures fabricated by laser interference with subsequent etching" dealing with structure sizes of similar dimensions and published in Nanomaterials 10(7) 1313 (2020), doi: 10.3390/nano10071313 .